# Seroprevalence of hospital staff in a province with zero COVID-19 cases

**Tanawin Nopsopon**[1], **Krit Pongpirul**[1,2,3]*, **Korn Chotirosniramit**[1], **Wutichai Jakaew**[4], **Chuenkhwan Kaewwijit**[4], **Sawan Kanchana**[4¤], **Narin Hiransuthikul**[1]

1 Department of Preventive and Social Medicine, Faculty of Medicine, Chulalongkorn University, Bangkok, Thailand, 2 Department of International Health, Johns Hopkins Bloomberg School of Public Health, Baltimore, Maryland, United States of America, 3 Bumrungrad International Hospital, Bangkok, Thailand, 4 Ranong Hospital, Ranong, Thailand

¤ Current address: Chumphon Khet Udomsakdi Hospital, Chumphon, Thailand
* doctorkrit@gmail.com

## Abstract

### Background

COVID-19 seroprevalence data, particularly in less developed countries with a relatively low incidence, has been scant. We aimed to explore the seroprevalence of hospital staff in the area with zero confirmed COVID-19 case to shed light on the situation of COVID-19 infection in zero or low infection rate countries where mass screening was not readily available.

### Methods

A locally developed rapid immunoglobulin M (IgM)/immunoglobulin G (IgG) test kit was used for hospital staff screening of Ranong hospital which is located in a province with zero COVID-19 prevalence in Thailand from 17th April to 17th May 2020. All staff was tested, 100 of which were randomly invited to have a repeating antibody test in one month. (Thai Clinical Trials Registry: TCTR20200426002)

### Results

Of 844 hospital staff, 82 were tested twice one month apart (response rate for repeating antibody test 82%). Overall, 0.8% of the participants (7 of 844) had positive IgM, none had positive IgG. Female staff had 1.0% positive IgM (95% CI: 0.5–2.1%) while male had 0.5% positive IgM (95% CI: 0.1–2.6%). No participants with a history of travel to the high-risk area or close contact with PCR-confirmed COVID-19 case developed SARS-CoV-2 antibodies. Among 844 staff, 811 had no symptoms and six of them developed IgM seropositive (0.7%) while 33 had minor symptoms and only one of them developed IgM seropositive (3.0%). No association between SARS-CoV-2 IgM status and gender, history of travel to a high-risk area, close contact with PCR-confirmed or suspected COVID-19 case, presence of symptoms within 14 days, or previous PCR status was found. None of the hospital staff developed SARS-CoV-2 IgG.

**Data Availability Statement:** All relevant data are within the manuscript and its Supporting Information files.

**Funding:** The author(s) received no specific funding for this work.

**Competing interests:** The authors have declared that no competing interests exist.

## Conclusions

COVID-19 antibody test could detect a considerable number of hospital staff who could be potential silent spreaders in a province with zero COVID-19 cases. Accurate antibody testing is a valuable screening tool, particularly in asymptomatic healthcare workers.

**Trial registration:** This study was approved by the Institutional Review Board of Chulalongkorn University (IRB No.236/63) and the Institutional Review Board of Ranong Hospital. (Thai Clinical Trials Registry: TCTR20200426002).

## Introduction

Seroprevalence data has been scarce in Asian countries besides China. Along with the gold-standard polymerase chain reaction (PCR) testing, antibody testing is beneficial for epidemiological investigation and epidemic control of infectious diseases including the present coronavirus disease 2019 (COVID-19). In Singapore, serological evidence was used to trace and identify missing spreaders for three clusters [1]. Asymptomatic silent spreaders have been of major concern as suggested by a systematic review—could be as low as 1.6% of COVID-19 confirmed cases in China or as high as 51.7% of confirmed cases in Diamond Princess cruise [2]. An estimated proportion of asymptomatic COVID-19 patients was 13.34% and was substantially higher in healthcare workers at 36.96% [3] with the transmission rate of asymptomatic patients to close contact individuals at 18.8% [4].

Early COVID-19 prevalence studies were based only on PCR for the diagnosis of severe acute respiratory syndrome coronavirus 2 (SARS-CoV-2) infection in individuals. However, recent studies tended to report PCR along with serological test results. An early report of an overall seroprevalence of 2.5% in a hospital setting in China, in which 1.8% and 3.5% were among healthcare workers and asymptomatic patients, respectively [5]. The recent meta-analysis of evidence up to 24[th] August 2020 reported an estimated overall seroprevalence in healthcare workers at 8.7% [6]. Additionally, China studied the development of antibodies against SARS-CoV-2 in symptomatic confirmed COVID-19 cases and found that immunoglobulin M (IgM) reached its peak 20–22 days after onset while immunoglobulin G (IgG) reached its peak 17–19 days after onset [7]. A more recent systematic review reported IgM median seroconversion time ranged from four to 14 days, peak at two to five weeks, and undetectable at six weeks after onset while IgG median seroconversion time range from 12–15 days, peak at three to seven weeks, and decline after eight weeks post-onset [8]. Some works emphasized antibody testing for the hospital workforce and policy issues. Another seroprevalence study in Belgium conducted on healthcare personnel who worked in a tertiary hospital found 6.4% seroprevalence and identified some risk factors for developing antibodies against SARS-CoV-2 [9].

There were inter-regional variations of seroprevalence in healthcare personnel with an estimated seroprevalence of 12.7% in North America, 8.5% in Europe, 8.2% in Africa, and 4.0% in Asia [6] with a range of seroprevalence among healthcare workers from 0% in Alzintan, Libya [10] to 45.3% in London, UK [11]. Moreover, intra-regional variations of seroprevalence were observed in healthcare workers, for example, seroprevalence among hospital staff in China range from 1.8% [5] to 17.1% [12] whereas seroprevalence of healthcare workers in Asia besides China and Southeast Asia ranged from 0.4% [13] to 9.1% [14]. In Southeast Asia, there were only two studies on the seroprevalence of COVID-19 in healthcare workers: one from Thailand reported a 3.7% seroprevalence [15], and one from Vietnam reported a zero seroprevalence [16]. Seroprevalences in healthcare personnel varied from 1.3% [17] to 45.3% [11]

in Europe, from 0.8% [18] to 44.0% [19] in North America, from 0.9% [20] to 14.1% [21] in South America, and from zero [10] to 45.1% [22] in Africa whereas no seroprevalence studies were from Antarctica or Australia.

Ideally, both PCR and antibody testing provided complementing information to shape the picture of the situation in a specific hospital, area, or country. However, most low- to middle-income countries could not afford the cost of laboratory tests and had to develop criteria-based policies for resource-use optimization. In Thailand, for instance, the PCR is reserved for individuals who meet the national criteria for COVID-19 PCR testing.

Ranong is one of 77 provinces in Thailand with zero cumulative confirmed COVID-19 cases from 17th April to 17th May 2020, and still had no case as of December 5, 2020 (Fig 1). Hospital is one of the highest risk areas for receiving and spreading pathogens—healthcare workers developed a higher chance of getting infected by co-workers or patients and vice versa. This study aims to estimate the hospital-wide seroprevalence in healthcare workers who worked in the largest public hospital in Ranong to develop the strategies to slow down the pandemic and ensure the safety of healthcare workers who come to work and patients who visit the hospital.

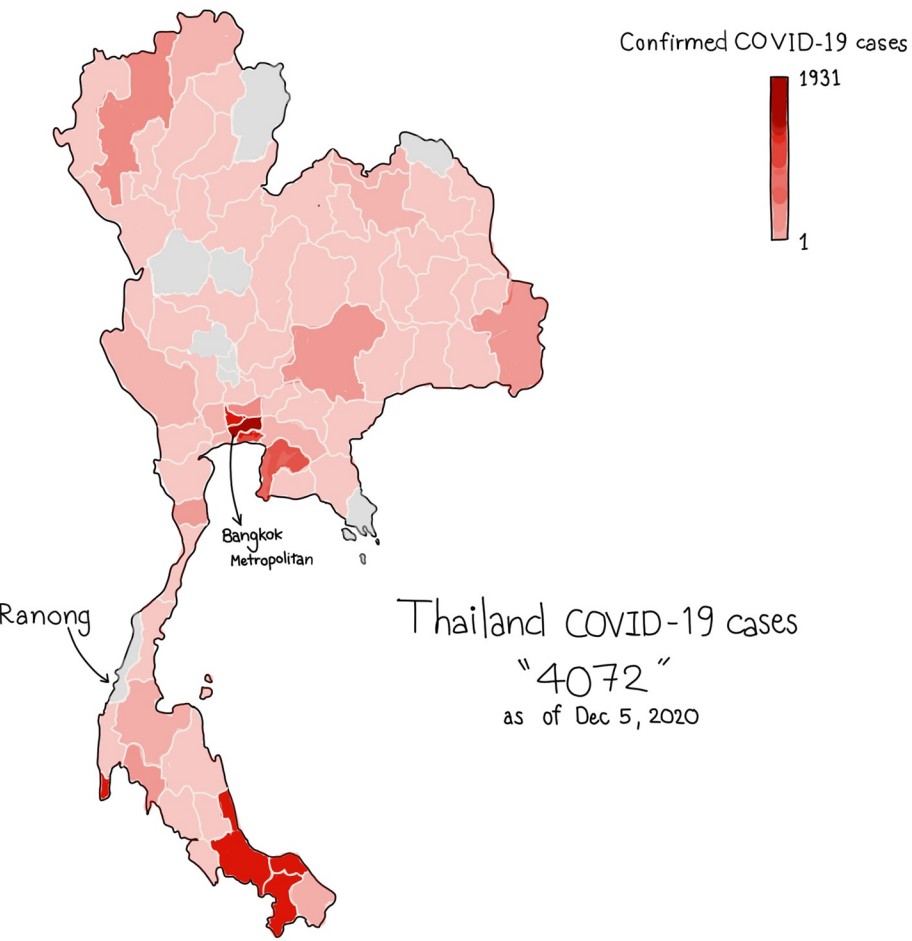

**Fig 1. Numbers of confirmed COVID-19 cases across geographical areas of Thailand as of July 4, 2020.** Bangkok metropolitan had the most cumulative confirmed COVID-19 cases, 1,609 cases, while Ranong, the province of focus in this study, had no confirmed COVID-19 case.

## Methods

### Participant

All 844 healthcare workers including the physician, nurse, medical assistance, medical technician, and non-medical officers of the largest public hospital in Ranong province that reported zero cumulative PCR-confirmed COVID-19 cases during the entire study period (17[th] April to 17[th] May 2020), were invited to participate in this study. All of them accepted to participate with written informed consent. Participants with active symptoms suiting national criteria for polymerase chain reaction testing were quarantined and excluded. Participants were asked to answer a survey about risk history for COVID-19, recent symptoms in the past two weeks, and previous PCR tests if available.

### Antibody testing

The locally developed Baiya Rapid IgG/IgM test kit (Baiya Phytopharm, Thailand) which reported the presence of IgM and IgG qualitatively, was used for antibody testing in this study. The internal validation of the test kit using the serum of 51 PCR confirmed COVID-19 cases and 150 controls showed sensitivity 94.1% (48 of 51) and specificity 98.0% (147 of 150) of IgM or IgG antibody. Participants with positive IgM were encouraged to have a PCR test if available.

### Study procedures

On 17[th] April 2020, of 844 participants, 100 were randomly selected to have their first antibody testing. On 17[th] May 2020, all the remaining participants were tested for serological immunity whereas the 100 healthcare workers who tested in April had their second antibody testing.

### Statistical analysis

Categorical data were presented with counts and percentages while continuous data were reported with median and interquartile range. Association between categorical variables and the status of immunoglobulin was analyzed using Fisher's exact test. The 95% confidence interval (CI) of the seroprevalence was calculated by Wilson's method using binomial probabilities. Missing data were excluded. All data were analyzed using Stata 16.1 (College Station, TX).

### Ethics committee approval

This study was approved by the Institutional Review Board of Chulalongkorn University (IRB No.236/63) and the Institutional Review Board of Ranong Hospital. (Thai Clinical Trials Registry: TCTR20200426002)

### Patient and public involvement

Patients and the public have not been directly involved in the design, conduct, or reporting of this study. However, we believe that our study would provide additional information for patients who would visit the hospital in zero or relatively low COVID-19 prevalence area and for the public to have more accessible real-world experience.

## Results

### Healthcare workers' demographic

All 844 Thai hospital staff were invited to participate in the study and tested for IgM and IgG antibodies against SARS-CoV-2 (100% participation rate). Their median age was 42 years

**Table 1. Demographic and clinical characteristics of Ranong hospital staff.**

|  | Total | IgM+ | |
|---|---|---|---|
|  |  | n (%) | 95% CI |
| Total | 844 | 7 (0.8%) | 0.4–1.7 |
| Median age, years (25th–75th percentile) | 42 (32–50) |  |  |
| Gender |  |  |  |
| Male, n (%) | 211 (25.0%) | 1 (0.5%) | 0.1–2.6 |
| Female, n (%) | 605 (71.7%) | 6 (1.0%) | 0.5–2.1 |
| Unspecified, n (%) | 28 (3.3%) | 0 (0.0%) | NA |
| Ethnicity |  |  |  |
| Thai | 844 (100.0%) | 7 (0.8%) | 0.4–1.7 |
| Non-Thai | 0 (0.0%) | 0 (0.0%) | NA |
| History of travel to high risk area |  |  |  |
| Yes | 21 (2.5%) | 0 (0.0%) | NA |
| No | 823 (97.5%) | 7 (0.9%) | 0.4–1.7 |
| History of close contact confirmed case |  |  |  |
| Yes | 17 (2.0%) | 0 (0.0%) | NA |
| No | 827 (98.0%) | 7 (0.8%) | 0.4–1.7 |
| History of close contact suspected case |  |  |  |
| Yes | 322 (38.1%) | 3 (0.9%) | 0.3–2.7 |
| No | 522 (61.9%) | 4 (0.8%) | 0.3–2.0 |
| Asymptomatic | 811 (96.1%) | 6 (0.7%) | 0.3–1.6 |
| Symptomatic | 33 (3.9%) | 1 (3.0%) | 0.5–15.3 |
| Fever | 2 (0.2%) | 0 (0.0%) | NA |
| Cough | 14 (1.7%) | 0 (0.0%) | NA |
| Rhinitis | 13 (1.5%) | 0 (0.0%) | NA |
| Sore throat | 12 (1.4%) | 1 (8.3%) | 1.5–35.4 |
| Dyspnea | 4 (0.5%) | 0 (0.0%) | NA |
| Previous PCR status |  |  |  |
| Negative | 8 (1.0%) | 0 (0.0%) | NA |
| Never tested | 836 (99.0%) | 7 (0.8%) | 0.4–1.7 |

Data were presented in counts and percentages unless otherwise specified.

IgM+, immunoglobulin M positive; NA, not available; PCR, polymerase chain reaction.

(interquartile range 32–50). Most of them (71.7%) were female and 96.1% had no symptoms. The 33 symptomatic participants (3.9%) reported cough (1.7%), rhinitis (1.5%), sore throat (1.4%), dyspnea (0.5%), and fever (0.2%). History of travel to the high-risk area was 2.5%, history of close contact PCR confirmed COVID-19 case was 2.0%, history of close contact suspected case was 38.1%. Only 1% of participants had previous negative PCR results while the rest never got tested (Table 1).

## Serological results of healthcare workers in Ranong hospital

Overall, seven hospital staff tested positive for COVID-19 IgM (0.8%, 95% CI: 0.4–1.7%) while none of the participants developed IgG. Female staff had 1.0% positive IgM (95% CI: 0.5–2.1%) while male staff had 0.5% positive IgM (95% CI: 0.1–2.6%). None of the participants with a history of travel to the high-risk area or a history of close contact with PCR-confirmed COVID-19 case developed antibodies against SARS-CoV-2. There was no statistical difference in IgM seroprevalence between staff with and without a history of close contact with suspected

COVID-19 cases. Among 844 staff, 811 had no symptoms and six of them developed IgM sero-positive (0.7%, 95% CI: 0.3–1.6%) while 33 had minor symptoms and only one of them developed immunoglobulin M (3.0%, 95% CI: 0.5–15.3%). Of 12 staff with a sore throat, one had positive IgM (8.3%, 95% CI: 1.5–35.4%). There was zero IgM seroprevalence in staff with fever, cough, rhinitis, or dyspnea. Of 844 participants, eight had previous negative PCR results and none has developed the antibody for SARS-CoV-2 (Table 1). No association between IgM antibody against SARS-CoV-2 status and gender, history of travel to a high-risk area, history of close contact with PCR-confirmed COVID-19 case, history of close contact with suspected COVID-19 case, presence of symptoms within 14 days, or previous PCR status was found. None of the hospital staff developed IgG against SARS-CoV-2.

## Characteristics of seropositive participants

Seven participants developed IgM antibodies in May. Their age ranged from 20 to 49 years. Six of them were female while only one staff was male. All of them were Thai, had no history of travel to a high-risk area, no history of contact with PCR-confirmed COVID-19 case, and no previous PCR status. Three had a history of contact with suspected COVID-19 cases while the other four did not. Participant 7 was the only one with IgM positive who had a symptom (i.e. sore throat) within 14 days before antibody testing. All seven staff with a positive antibody subsequently got nasopharyngeal swabs for PCR and resulted in negative for SARS-CoV-2 on PCR (Table 2).

## Repeating antibody tests in random sampling participants

Of 844 participants, 100 were randomly selected to get antibody testing in April and then repeat in May at hospital-wide antibody testing. None of them had any immunoglobulin developed in April. Of 100 randomly selected participants, 82 participated in antibody testing in May (response rate for repeating antibody test 82%) which showed no antibody against COVID-19.

## Discussion

In a small seaside province, Ranong, which had 193,370 inhabitants, located in the south of Thailand closed to the Andaman Sea and had a border close to Myanmar, there was zero

**Table 2. Characteristics of hospital staff who developed immunoglobulin M antibody and subsequent PCR status.**

| | Age Range, years | Gender | Ethnicity | History of travel | History of contact with a confirmed case | History of contact with a suspected case | Previous PCR status | Symptoms | Subsequent PCR status |
|---|---|---|---|---|---|---|---|---|---|
| Participant 1 | 30–39 | Female | Thai | No | No | Yes | Never tested | No | Negative |
| Participant 2 | 20–29 | Female | Thai | No | No | No | Never tested | No | Negative |
| Participant 3 | 40–49 | Female | Thai | No | No | No | Never tested | No | Negative |
| Participant 4 | 40–49 | Female | Thai | No | No | Yes | Never tested | No | Negative |
| Participant 5 | 30–39 | Female | Thai | No | No | No | Never tested | No | Negative |
| Participant 6 | 40–49 | Male | Thai | No | No | No | Never tested | No | Negative |
| Participant 7 | 40–49 | Female | Thai | No | No | Yes | Never tested | Sore throat | Negative |

PCR, polymerase chain reaction.

officially PCR-confirmed COVID-19 case compared to the current situation in Thailand (0.048 per thousand). This zero prevalence could be either from good compliance to public health recommendations or a low number of individuals who get PCR testing due to the stringent national criteria.

In this study, we reported a 0.8% IgM seroprevalence in Ranong hospital staff while their PCR tests were negative. While critics might argue that the antibody test had a high false-positive rate, especially in a population with high pre-test probability such as hospital staff, the PCR test could have a high false-negative rate, especially without a highly sensitive diagnostic test [23].

Of seven seropositive hospital staff, only one was symptomatic and none of them had a history of travel to high-risk areas or history of close contact with PCR-confirmed COVID-19 case. The current national criteria and policies for PCR testing mostly rely on risk histories and symptoms so asymptomatic individuals or those with minor symptoms were left out, resulting in an underestimation of the actual prevalence of COVID-19. However, the lack of association with risk history might be because of the small number of staff with positive IgM.

Our study reported a lower seroprevalence in healthcare workers than hospitals in China (0.8% vs. 1.8%) [5], a tertiary hospital in Belgium (0.8% vs. 6.4%) [9], and a rural hospital located in low COVID-19 prevalence county of Germany (0.8% vs 2.7%) [24]. Unlike China, Belgium, and Germany where the seroprevalences were mostly from positive IgG, our study revealed mostly positive IgM. Comparison with Belgium and German hospitals should be interpreted with caution due to the unknown PCR status of subjects of the two studies.

All seropositive healthcare workers in our study had positive IgM and negative IgG. We speculated two possible explanations that might associate with the negative IgG result. First, the time frame of antibody testing from the onset of the disease might play a crucial role [25]. While IgM and IgG can be detected in some patients as early as 4–6 and 5–10 days after symptom onset, respectively [26], 70% of symptomatic patients developed positive IgM by days 8–14 and 90% of total antibody tests positive by days 11–24 [27]. Moreover, COVID-19 confirmed patients had higher SARS-CoV-2 IgM concentration than IgG before the 15th-day post symptom onset and vice versa after the 15th-day post symptom onset [28], and patients developed peak IgM of 94.1% approximately 20–22 days and all of them finally have positive IgG approximately 17–19 days after symptom onset [7]. Thus, antibody testing during the window period could lead to a false-negative result of IgG antibody [29]. For patients with no or minor symptoms, the onset might not be possible to determine. Moreover, IgM can be persisted for 42 days post symptom [30]. Nonetheless, the time of antibody detection might not reflect the time of infection and disease transmission. Secondly, the difference in patients' immune response might be a relevant factor. Some patients recover from COVID-19 infection without any production of SAR-CoV-2 IgG, reflecting that strong innate immunity might be sufficient to eradicate SARS-CoV-2 [31].

In this study, we used a qualitative antibody test kit so the relatively lower antibody level in early infected persons might not be detectable. Repeating PCR or quantitative antibody tests for suspected individuals was practiced in many countries. However, the cost would be doubled or more, and might not be possible for low- or middle-income countries to cover all suspected individuals. We prospectively designed the workflow to get participants with positive IgM, which related to acute infection of SARS-CoV-2, subsequently tested with a nasopharyngeal swab for PCR which was the gold standard for diagnosis COVID-19 during the study period. We also attempted to repeat the antibody test by prospectively randomly selected 100 participants to get antibody testing in April and then participate in hospital-wide antibody testing in May for repeating test; however, only 82 of them could participate in the follow-up test and none of them developed antibodies against SARS-CoV-2 in both first and second

tests. All IgM positive participants were tested in hospital-wide antibody testing in May and we did not have a chance to repeat the antibody test in seropositive participants; however, all of them were tested for PCR after IgM positive as planned. Rapid antibody tests should be more affordable to permit multiple tests among high-risk asymptomatic individuals who could be silent spreaders.

While the data is relatively limited, this data is the best evidence we can obtain in the current situation in Thailand where the appropriate COVID-19 test in population has not been conducted. Moreover, as the hospital underwent a major significant change in its organizational structure and management team, along with the compliance to the ethical approval we received, our request to conduct repeat surveillance of the same cohort has not been granted.

Serological testing provides some crucial epidemiological information and would have been more effective when combined with other diagnostic tests such as PCR. With immunoglobulin status and PCR results, we can shape the situation more precisely for both individual and regional views. The antibody testing should be used as a screening tool, not a diagnostic tool, and should be conducted with consideration of the current situation in each area. Hopefully, with this and other vigorous and dedicated studies on antibody status around the globe, antibody testing would provide useful information for pandemic control.

## Conclusions

The COVID-19 antibody test could detect a considerable number of hospital staff who could be potential silent spreaders in areas with zero COVID-19 cases. Accurate antibody testing is a valuable screening tool, particularly in asymptomatic healthcare workers.

## Supporting information

**S1 Data.**
(XLSX)

## Acknowledgments

We thank the staff of Ranong hospital for their kind participation in this study and Dr. Irin Lertparinyaphorn for preparing the figure.

## Author Contributions

**Conceptualization:** Tanawin Nopsopon, Krit Pongpirul, Sawan Kanchana, Narin Hiransuthikul.

**Data curation:** Tanawin Nopsopon, Krit Pongpirul, Korn Chotirosniramit, Wutichai Jakaew.

**Formal analysis:** Tanawin Nopsopon.

**Investigation:** Krit Pongpirul, Korn Chotirosniramit, Wutichai Jakaew, Chuenkhwan Kaewwijit.

**Methodology:** Tanawin Nopsopon, Krit Pongpirul, Narin Hiransuthikul.

**Project administration:** Krit Pongpirul, Wutichai Jakaew, Sawan Kanchana.

**Resources:** Krit Pongpirul, Sawan Kanchana, Narin Hiransuthikul.

**Software:** Korn Chotirosniramit.

**Supervision:** Krit Pongpirul, Wutichai Jakaew, Sawan Kanchana, Narin Hiransuthikul.

**Validation:** Tanawin Nopsopon, Krit Pongpirul, Wutichai Jakaew, Chuenkhwan Kaewwijit.

**Visualization:** Tanawin Nopsopon, Korn Chotirosniramit.

**Writing – original draft:** Tanawin Nopsopon, Krit Pongpirul.

**Writing – review & editing:** Tanawin Nopsopon, Krit Pongpirul, Korn Chotirosniramit, Wutichai Jakaew, Chuenkhwan Kaewwijit, Sawan Kanchana, Narin Hiransuthikul.

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
