## [Decision Letter · Decision Letter 0]

17 Nov 2020

PONE-D-20-25321

Seroprevalence of Hospital Staff in Province with Zero COVID-19 Cases

PLOS ONE

Dear Dr. Krit Pongpirul,

Thank you for submitting your manuscript to PLOS ONE. After careful consideration, we feel that it has merit but does not fully meet PLOS ONE’s publication criteria as it currently stands. The data presented is very limited (only a one month period). More recent surveillance of the same cohort will be preferable and compared to the results obtained from the earlier time point. Therefore, we invite you to submit a revised version of the manuscript that addresses all the points raised during the review process.

We look forward to receiving your revised manuscript.

Kind regards,

Daniela Flavia Hozbor

Academic Editor

PLOS ONE

Journal Requirements:

2.We note that [Figure(s) 1] in your submission contain map images which may be copyrighted. All PLOS content is published under the Creative Commons Attribution License (CC BY 4.0), which means that the manuscript, images, and Supporting Information files will be freely available online, and any third party is permitted to access, download, copy, distribute, and use these materials in any way, even commercially, with proper attribution. For these reasons, we cannot publish previously copyrighted maps or satellite images created using proprietary data, such as Google software (Google Maps, Street View, and Earth). For more information, see our copyright guidelines: http://journals.plos.org/plosone/s/licenses-and-copyright.

1.    You may seek permission from the original copyright holder of Figure(s) [1] to publish the content specifically under the CC BY 4.0 license. 

3.Thank you for stating the following in the Acknowledgments Section of your manuscript:

[We thank Baiya Phytopharm, Thailand for supporting the Baiya Rapid COVID-19 IgG/IgM test kit. The

211 company did not involve in the data analysis, interpretation, or manuscript preparation.]

 [The author(s) received no specific funding for this work.]

Additionally, because some of your funding information pertains to [commercial funding//patents], we ask you to provide an updated Competing Interests statement, declaring all sources of commercial funding.

In your Competing Interests statement, please confirm that your commercial funding does not alter your adherence to PLOS ONE Editorial policies and criteria by including the following statement: "This does not alter our adherence to PLOS ONE policies on sharing data and materials.” as detailed online in our guide for authors  http://journals.plos.org/plosone/s/competing-interests.  If this statement is not true and your adherence to PLOS policies on sharing data and materials is altered, please explain how.

Please include the updated Competing Interests Statement and Funding Statement in your cover letter. We will change the online submission form on your behalf.

4.We noticed you have some minor occurrence of overlapping text with the following previous publication(s), which needs to be addressed: https://www.medrxiv.org/content/10.1101/2020.06.24.20139188v1?versioned=true

In your revision ensure you cite all your sources (including your own works), and quote or rephrase any duplicated text outside the methods section. Further consideration is dependent on these concerns being addressed.

Reviewers' comments:

Reviewer's Responses to Questions

**Comments to the Author**

1. Is the manuscript technically sound, and do the data support the conclusions?

Reviewer #1: Yes

Reviewer #2: Partly

2. Has the statistical analysis been performed appropriately and rigorously? 

Reviewer #1: Yes

Reviewer #2: Yes

3. Have the authors made all data underlying the findings in their manuscript fully available?

Reviewer #1: Yes

Reviewer #2: Yes

4. Is the manuscript presented in an intelligible fashion and written in standard English?

Reviewer #1: Yes

Reviewer #2: Yes

5. Review Comments to the Author

Reviewer #1: This manuscript by Nopsopon et al described the SARS-CoV-2 seroprevalence of the hospital staff of one of the largest hospitals in the Ranong province of Thailand, where no confirmed COVID-19 was found by between April-May, 2020. The testing was carried out using a homemade IgM/IgG detection test. The authors found that among the 844 members tested only 7 cases were found to be positive for IgM but negative for IgG. Only 1 of the positive cases has reported sore throat. All the positive cases were PCR negative. This is the first report of the testing of hospital staff in the region, however, based on numerous reports it is hardly surprising that there are individuals who have prior exposure to the virus but have yet remained asymptomatic. The data presented is very limited and for only a one month period, way back in April-May. With the fast changing pandemic more recent surveillance of the same cohort will be preferable and compared to the results obtained from the earlier time point. The addition data will substantially strengthen the manuscript. In addition, there are several other concerns.

1. It is not clear why only IgM and not IgG was detected. This is quite surprising. This needs to be followed up and explain whether this could be due to experimental errors. In addition, were the cases being followed up and retested for Ab and for IgG?

2. There is a concern on the test sensitivity of 94.1% (48 of 51), some positive cases could have been missed.

3. Was second Ab tested carried out with the IgM positive cases?

4. It is not clear whether any hospital staff were excluded. It was indicated in line 106 that “Participants with active symptoms suiting national criteria for polymerase chain reaction testing were quarantined and excluded.

5. It is not clear whether 84 or 100 individuals were tested twice, conflicting numbers were provided in the abstract versus the text.

Reviewer #2: Burden of disease deals with a range of medical statistics aimed to give a comprehensive picture of how diseases impact on society or to allocate health care and research resources. Diagnostic tests are an important part in this regard. A perfect test, usually referred to as the gold standard implies being positive in all patients with the disease and negative in all patients who do not have the disease. However, most tests are imperfect. In the work by Pongpirul et al. the authors carried out a survey assessing the presence of anti-COVID-19 antibodies in hospital workers from a Thailand area in which COVID-19 had not been reported during the study period. Among 844 participants few of them (<1%) revealed the presence of IgM antibodies bearing no close relationship with their clinical or personal characteristics. The study adds some information to the local features of COVID-19 infection.

It would be interesting to know whether there existed a different degree of exposure risk among staff members.

It is intriguing that people yielding an IgM positive test at the beginning failed to develop IgG response a month later. Any tentative explanation for this?

Information is needed on whether validation studies were carried in recent COVID-19 cases, or not?

The same applies when talking about the amount of missing data

In discussing results please note that lack of association may depend on the reduced number of people yielding a positive IgM response

Page 8, lines 191-192, there are more studies in this regard which should be mentioned.

Please also note that the time when Ab are detected may have nothing to do with transmission

The Bangkok Metropolitan area might be indicated in the Figure.

6. PLOS authors have the option to publish the peer review history of their article (what does this mean?). If published, this will include your full peer review and any attached files.

Reviewer #1: No

Reviewer #2: No

---

## [Author Response · Author response to Decision Letter 0]

8 Dec 2020

Dear Editor,

We thank you and the reviewers for the comments and suggestions. Please find our point-by-point responses below:

Editor: Thank you for submitting your manuscript to PLOS ONE. After careful consideration, we feel that it has merit but does not fully meet PLOS ONE’s publication criteria as it currently stands. The data presented is very limited (only a one month period). More recent surveillance of the same cohort will be preferable and compared to the results obtained from the earlier time point. Therefore, we invite you to submit a revised version of the manuscript that addresses all the points raised during the review process.

Response: Thank you for your time and consideration. The manuscript was revised to meet PLOS ONE’s publication criteria. While the data is relatively limited, this data is the best evidence we can obtain in the current situation in Thailand where the appropriate COVID-19 test in population has not been conducted. Moreover, as the hospital underwent a major significant change in its organizational structure and management team, along with the compliance to the ethical approval we initially received, our request to conduct repeat surveillance of the same cohort has not been granted yet and might take several months. Nonetheless, please note that, as of December 7, 2020, this province still reports zero cases. We added these limitation statements to the Discussion section accordingly.

Editor: 2.We note that [Figure(s) 1] in your submission contain map images which may be copyrighted. All PLOS content is published under the Creative Commons Attribution License (CC BY 4.0), which means that the manuscript, images, and Supporting Information files will be freely available online, and any third party is permitted to access, download, copy, distribute, and use these materials in any way, even commercially, with proper attribution. For these reasons, we cannot publish previously copyrighted maps or satellite images created using proprietary data, such as Google software (Google Maps, Street View, and Earth).

Response: As Figure 1 contains original data from our research, we provide a replacement figure with an original image from our colleague whom we acknowledged in the revised manuscript so the copyright issue should no longer be a concern. As of December 7, 2020, this province still reports zero cases so we updated the data in Figure 1 as well.

Editor: 3.Thank you for stating the following in the Acknowledgments Section of your manuscript:

[We thank Baiya Phytopharm, Thailand for supporting the Baiya Rapid COVID-19 IgG/IgM test kit. The company did not involve in the data analysis, interpretation, or manuscript preparation.]

[The author(s) received no specific funding for this work.]

Additionally, because some of your funding information pertains to [commercial funding//patents], we ask you to provide an updated Competing Interests statement, declaring all sources of commercial funding.

Response: We did not mean to provide funding information in the Acknowledgment Section. None of the authors receive any funding from the company mentioned in the Acknowledgment Section. We only wanted to thank them for their contribution to developing the local COVID-19 IgG/IgM test kit for Thailand upon request free of charge, which resulted in an opportunity for us to conduct this observational study. We revised the choice of words in the Acknowledgment Section accordingly.

Editor: In your Competing Interests statement, please confirm that your commercial funding does not alter your adherence to PLOS ONE Editorial policies and criteria by including the following statement: "This does not alter our adherence to PLOS ONE policies on sharing data and materials.” as detailed online in our guide for authors http://journals.plos.org/plosone/s/competing-interests. If this statement is not true and your adherence to PLOS policies on sharing data and materials is altered, please explain how. Please include the updated Competing Interests Statement and Funding Statement in your cover letter. We will change the online submission form on your behalf.

Response: Authors have no competing interests according to the PLOS definition. The suggested statement is added in the Competing Interests statement. The updated Competing Interests Statement and Funding Statement were included in the cover letter.

Editor: 4.We noticed you have some minor occurrence of overlapping text with the following previous publication(s), which needs to be addressed: https://www.medrxiv.org/content/10.1101/2020.06.24.20139188v1?versioned=true

In your revision ensure you cite all your sources (including your own works), and quote or rephrase any duplicated text outside the methods section. Further consideration is dependent on these concerns being addressed.

Response: All sources used in this manuscript are already cited in the previously submitted manuscript (including the concerning publication mentioned above – Ref. 12). All overlapping text outside the methods section is rephrased. Please advise if we still have to address this concern differently.

Reviewer #1: This manuscript by Nopsopon et al described the SARS-CoV-2 seroprevalence of the hospital staff of one of the largest hospitals in the Ranong province of Thailand, where no confirmed COVID-19 was found by between April-May, 2020. The testing was carried out using a homemade IgM/IgG detection test. The authors found that among the 844 members tested only 7 cases were found to be positive for IgM but negative for IgG. Only 1 of the positive cases has reported sore throat. All the positive cases were PCR negative. This is the first report of the testing of hospital staff in the region, however, based on numerous reports it is hardly surprising that there are individuals who have prior exposure to the virus but have yet remained asymptomatic. The data presented is very limited and for only a one month period, way back in April-May. With the fast changing pandemic more recent surveillance of the same cohort will be preferable and compared to the results obtained from the earlier time point. The addition data will substantially strengthen the manuscript. In addition, there are several other concerns.

Response: Thank you for your time in reviewing the manuscript. Despite the limitation of data in this manuscript, we provide the best evidence available in the current situation in Thailand where COVID-19 testing is low, given the limited resources and national screening policy. With the volatile nature of the pandemic, we would like to provide evidence of an actual situation where the zero or low number of COVID-19 confirmed cases in Thailand might come from a low number of people tested. Although we wanted to conduct a repeat survey as suggested, the hospital underwent a major significant change in its organizational structure and management team, along with the compliance to the ethical approval we received; hence, our request to conduct repeat surveillance of the same cohort has not been granted and might take several months. We added these limitation statements to the Discussion section accordingly (Line 194-198). We believe that the data we provided would bring attention to the overlooked problem of the low number of testing in this region, thus encourage the governing body to take more action and change the perspective that only high-risk or symptomatic patients should get tested for mass screening to get the actual situation in Thailand.

Reviewer #1: 1. It is not clear why only IgM and not IgG was detected. This is quite surprising. This needs to be followed up and explain whether this could be due to experimental errors. In addition, were the cases being followed up and retested for Ab and for IgG?

Response: More discussion on predominant IgM positive compared to IgG positive is provided in Discussion section. Mass screening or antibody testing has not been readily supported in Thailand, especially in the current situation where the national policy allows people to get tested as low as possible to keep the low number of confirmed cases. We really would like to conduct a follow-up test to provide the most recent situation in Thailand; however, without the first piece of evidence to demonstrate how mass screening would provide benefit and reality of COVID-19 situation in Thailand, the follow-up test is currently not possible. The cases were not retested for Antibody but instead get PCR testing which, at that time, was considered the gold standard for COVID-19 confirmation in Thailand.

Reviewer #1: 2. There is a concern on the test sensitivity of 94.1% (48 of 51), some positive cases could have been missed.

Response: We concern about some positive cases would be missed by this rapid IgM/IgG test kit we used; however, this is the best readily available test that could be used as mass screening in Thailand at the time we conducted the study. As of December 7, 2020, this province still reports zero cases.

Reviewer #1: 3. Was second Ab tested carried out with the IgM positive cases?

Response: The cases were not retested for Antibody but instead get PCR testing which, at that time, was considered the gold standard for COVID-19 confirmation in Thailand.

Reviewer #1: 4. It is not clear whether any hospital staff were excluded. It was indicated in line 106 that “Participants with active symptoms suiting national criteria for polymerase chain reaction testing were quarantined and excluded.

Response: We are sorry for the confusion and thank you very much for pointing that out. The statement “Participants with active symptoms suiting national criteria for polymerase chain reaction testing were quarantined and excluded.” was in the Methods section to describe how our observational study concurs with the current COVID-19 screening policy in Thailand. Hence, no staff was excluded from our study per se.

Reviewer #1: 5. It is not clear whether 82 or 100 individuals were tested twice, conflicting numbers were provided in the abstract versus the text.

Response: Again, we are sorry for the confusion. We revised both the “Repeating antibody tests in random sampling participants” subsection of the Method section as follows: “Of 844 participants, 100 were randomly selected to get repeating antibody testing in April and May. None of them had any immunoglobulin developed in April. Of 100 randomly selected participants, 82 participated in antibody testing in May (response rate for repeating antibody test 82%) which showed no antibody against COVID-19.” The Abstract has also been revised accordingly.

Reviewer #2: Burden of disease deals with a range of medical statistics aimed to give a comprehensive picture of how diseases impact on society or to allocate health care and research resources. Diagnostic tests are an important part in this regard. A perfect test, usually referred to as the gold standard implies being positive in all patients with the disease and negative in all patients who do not have the disease. However, most tests are imperfect. In the work by Pongpirul et al. the authors carried out a survey assessing the presence of anti-COVID-19 antibodies in hospital workers from a Thailand area in which COVID-19 had not been reported during the study period. Among 844 participants few of them (<1%) revealed the presence of IgM antibodies bearing no close relationship with their clinical or personal characteristics. The study adds some information to the local features of COVID-19 infection.

Response: Thank you very much. We are pleased to know that our study is useful.

Reviewer #2: It would be interesting to know whether there existed a different degree of exposure risk among staff members.

Response: Staff members consist of many occupations which have various degree of exposure risk, thus we provide the history of close contact confirmed COVID-19 case and close contact suspected COVID-19 case to assess association with a history of risk exposure and seroprevalence. 

Reviewer #2: It is intriguing that people yielding an IgM positive test at the beginning failed to develop IgG response a month later. Any tentative explanation for this?

Response: Staffs with IgM positive did not have an antibody test a month later, instead they have PCR testing which was considered the gold standard at that time. For repeating antibody testing, we randomly selected 100 from 844 staff in Ranong hospital to get antibody testing twice in April and May, which none of them developed antibodies against SARS-CoV-2.

Reviewer #2: Information is needed on whether validation studies were carried in recent COVID-19 cases, or not?

Response: Yes, the validation study for the rapid test kit was conducted in recent COVID-19 cases.

Reviewer #2: In discussing results please note that lack of association may depend on the reduced number of people yielding a positive IgM response.

Response: Thank you very much. The discussion about lack of association may depend on the reduced number of people yielding a positive IgM response is provided in the Discussion section as advised (Line 174-175).

Reviewer #2: Page 8, lines 191-192, there are more studies in this regard which should be mentioned.

Response: Additional studies are added as suggested (Line 182-193).

Reviewer #2: Please also note that the time when Ab are detected may have nothing to do with transmission.

Response: The discussion about the time when Ab is detected may have nothing to do with the transmission is provided in the Discussion Section.

Reviewer #2: The Bangkok Metropolitan area might be indicated in the Figure.

Response: The Bangkok Metropolitan area is now indicated in the revised Figure.

We hope that our responses are satisfactory. Should there be anything that might improve our work, please kindly inform us. Thank you very much for your kind consideration.

Best Regards,

Assist. Prof. Dr. Krit Pongpirul, MD, MPH, PhD.

On behalf of the authors

---

## [Decision Letter · Decision Letter 1]

4 Jan 2021

PONE-D-20-25321R1

Seroprevalence of Hospital Staff in Province with Zero COVID-19 Cases

PLOS ONE

Dear Dr. Krit Pongpirul,

Thank you for submitting your manuscript to PLOS ONE. After careful consideration, we feel that it has merit but does not fully meet PLOS ONE’s publication criteria as it currently stands. Therefore, we invite you to submit a revised version of the manuscript that addresses all the points raised during the review process.

We look forward to receiving your revised manuscript.

Kind regards,

Daniela Flavia Hozbor

Academic Editor

PLOS ONE

Reviewers' comments:

Reviewer's Responses to Questions

**Comments to the Author**

1. If the authors have adequately addressed your comments raised in a previous round of review and you feel that this manuscript is now acceptable for publication, you may indicate that here to bypass the “Comments to the Author” section, enter your conflict of interest statement in the “Confidential to Editor” section, and submit your "Accept" recommendation.

Reviewer #1: (No Response)

Reviewer #2: All comments have been addressed

2. Is the manuscript technically sound, and do the data support the conclusions?

Reviewer #1: Partly

Reviewer #2: Yes

3. Has the statistical analysis been performed appropriately and rigorously? 

Reviewer #1: Yes

Reviewer #2: Yes

4. Have the authors made all data underlying the findings in their manuscript fully available?

Reviewer #1: Yes

Reviewer #2: Yes

5. Is the manuscript presented in an intelligible fashion and written in standard English?

Reviewer #1: Yes

Reviewer #2: Yes

6. Review Comments to the Author

Reviewer #1: 1. The information in the introduction is outdated and needs to be updated, for example lines 61-64.

2. The weakness of the paper is the failing to resolve the cases with IgM and no IgG, with a second Ab test to resolve whether the test is missing these cases during very early phases of infection before the development of IgG or the problem is with the specificity of the assay. More interesting still is that there are infected cases that develop only IgM and no IgG, which has not yet been reported. The explanation by the authors that they were not the ones tested a second time is not satifactory. It is understandable that even though not everyone tested first round were retested a second time, but 100 out of them were retested a second time. It is very surprising that all seven IgM positive cases were not retested. The failure to retest the cases with IgM reflects a major weakness in the study design, and a scientific flaw.

3. The main point of the paper is to promote mass serological screening of the population, this recommendation should not be made given the concerns on the serology test itself and the detection of IgM only data.

Reviewer #2: Changes introduced into the manuscript fulfill with my comments and suggestions, for which the paper is now acceptable on my end

7. PLOS authors have the option to publish the peer review history of their article (what does this mean?). If published, this will include your full peer review and any attached files.

Reviewer #1: No

Reviewer #2: No

---

## [Author Response · Author response to Decision Letter 1]

19 Jan 2021

Dear Editor,

We thank you and the reviewers for the comments and suggestions. Please find our point-by-point responses below:

Editor: Thank you for submitting your manuscript to PLOS ONE. After careful consideration, we feel that it has merit but does not fully meet PLOS ONE’s publication criteria as it currently stands. Therefore, we invite you to submit a revised version of the manuscript that addresses all the points raised during the review process.

Response: Thank you for your time and consideration. The manuscript was revised to meet PLOS ONE’s publication criteria. The Introduction section was updated with more recent references. The concerned weakness was additionally discussed in detail in the Discussion section. The main point of the paper had been clarified and specified in the Discussion section.

Reviewer #1: 1. The information in the introduction is outdated and needs to be updated, for example lines 61-64.

Response: Thank you for your time and crucial comment. The information in the Introduction section was updated with more recent evidence published in late 2020 and Jan 2021 including original articles and systematic reviews. Some key information during early pandemics were preserved while unnecessary outdated information was deleted.

Reviewer #1: 2. The weakness of the paper is the failing to resolve the cases with IgM and no IgG, with a second Ab test to resolve whether the test is missing these cases during very early phases of infection before the development of IgG or the problem is with the specificity of the assay. More interesting still is that there are infected cases that develop only IgM and no IgG, which has not yet been reported. The explanation by the authors that they were not the ones tested a second time is not satifactory. It is understandable that even though not everyone tested first round were retested a second time, but 100 out of them were retested a second time. It is very surprising that all seven IgM positive cases were not retested. The failure to retest the cases with IgM reflects a major weakness in the study design, and a scientific flaw.

Response: The comments were well received; however, a seroprevalence study is usually cross-sectional. Several seroprevalence studies on COVID-19 performed only one antibody test in each asymptomatic individual (Xu 2020, Steensels 2020, Kammon 2020, Chen 2020, Nakamura 2020, Takita 2020, Nopsopon 2020, Chau 2020, Psichogiou 2020, Mughal 2020, Insúa 2020, Costa 2020, Olayanju 2020). For other types of study, multiple serological tests were performed in COVID-19 confirmed cases to assess the antibody dynamics (Long 2020, Post 2020, and Lee 2020). As our study attempted to get two consecutive antibody tests in some randomly selected asymptomatic individuals, along with the RT-PCR confirmation in the IgM positive participants that was permitted by the government, we believe that our study contributed relatively more information than a typical seroprevalence study and provided a real-life picture of the early COVID-19 pandemic.

Nonetheless, please allow us to clarify the design of our study as the description of testing sequences might have been confusing. The study randomly selected 100 individuals to get tested in April and then conducted hospital-wide antibody testing in May. According to the protocol approved by the institutional review board, a participant with positive IgM would get the nasopharyngeal swab for RT-PCR, which was gold standard during the study period, to confirm the diagnosis at the hospital expense whereas second antibody test for IgM positive patients was not in the approved protocol. All seven IgM positive participants were found in May, thus did not get their second antibody test. Given the limited test availability during the study period (early COVID-19 pandemic), the distribution and use of test kits were not totally under our control. Despite the importance of the second antibody test, we attempted to request to conduct the repeat surveillance but were not granted by the ethical committee due to the predilection of PCR testing according to the national policy mentioned above. Even if we could get the second antibody test among them, the sample size would still be too small to conclude in this regard. The manuscript was revised to clarify the concerning issue in the Result and Discussion sections.

Reviewer #1: 3. The main point of the paper is to promote mass serological screening of the population, this recommendation should not be made given the concerns on the serology test itself and the detection of IgM only data.

Response: Thank you for your comment. The message delivered in the Discussion section was revised to down tone the main point as advised.

Reviewer #2: Changes introduced into the manuscript fulfill with my comments and suggestions, for which the paper is now acceptable on my end

Response: Thank you for your time and great comments. The manuscript was drastically improved in quality through your precious advice.

We hope that our responses are satisfactory. Should there be anything that might improve our work, please kindly inform us. Thank you very much for your kind consideration.

Best Regards,

Assist. Prof. Dr. Krit Pongpirul, MD, MPH, PhD.

On behalf of the authors

---

## [Decision Letter · Decision Letter 2]

22 Feb 2021

PONE-D-20-25321R2

Seroprevalence of Hospital Staff in a Province with Zero COVID-19 Cases

PLOS ONE

Dear Dr. Krit Pongpirul,

Thank you for submitting your manuscript to PLOS ONE. After careful consideration, we feel that it has merit but does not fully meet PLOS ONE’s publication criteria as it currently stands. Therefore, we invite you to submit a revised version of the manuscript that addresses the points raised during the review process.

We look forward to receiving your revised manuscript.

Kind regards,

Daniela Flavia Hozbor

Academic Editor

PLOS ONE

Reviewers' comments:

Reviewer's Responses to Questions

**Comments to the Author**

1. If the authors have adequately addressed your comments raised in a previous round of review and you feel that this manuscript is now acceptable for publication, you may indicate that here to bypass the “Comments to the Author” section, enter your conflict of interest statement in the “Confidential to Editor” section, and submit your "Accept" recommendation.

Reviewer #1: All comments have been addressed

Reviewer #3: (No Response)

2. Is the manuscript technically sound, and do the data support the conclusions?

Reviewer #1: Partly

Reviewer #3: Yes

3. Has the statistical analysis been performed appropriately and rigorously? 

Reviewer #1: Yes

Reviewer #3: Yes

4. Have the authors made all data underlying the findings in their manuscript fully available?

Reviewer #1: Yes

Reviewer #3: Yes

5. Is the manuscript presented in an intelligible fashion and written in standard English?

Reviewer #1: Yes

Reviewer #3: Yes

6. Review Comments to the Author

Reviewer #1: The authors has addressed the concerns as best as they could without performing the requested additional testing results, which this reviewer still think is not satisfactory. Perhaps adding the rationale and explanation for having IgM and no IgG response in the tested case will provide additional scientific interest for the readers.

Reviewer #3: 1) Although I agree with the flaws/limitations of the study as summarized by Reviewer 1, the study does add informative descriptive data for this region in Thailand despite the limitations.

2) The statement that females have higher IgM % than males in Abstract and Results sections are not appropriate. Just provide the percentages and confidence intervals (CIs). As provided in the Results section, the CIs are nearly overlapping so it cannot be concluded that they are different. Please revise accordingly the statements below.

Abs: Females seemed to have higher IgM seropositive than male staff (1.0% vs. 0.5%).

Results: Female staff seemed to have higher rate of positive IgM (1.0%, 95% CI: 0.5–2.1%) than male (0.5%, 95% CI: 0.1–2.6%).

3) In the conclusion: Consider revising FROM:

“Antibody testing should be promoted for mass screening, particularly

in asymptomatic healthcare workers.”

TO:

“Accurate antibody testing is a valuable screening tool, particularly in

asymptomatic healthcare workers.”

4) Typo:

Page 7: Only 1% of participants had previous negative PCR results while the rest never get tested (Table 1).

… should be ‘got’ tested.

Please read the paper again carefully and correct any grammatical errors.

7. PLOS authors have the option to publish the peer review history of their article (what does this mean?). If published, this will include your full peer review and any attached files.

Reviewer #1: No

Reviewer #3: No

---

## [Author Response · Author response to Decision Letter 2]

26 Feb 2021

Dear Editor,

We thank you and the reviewers for the comments and suggestions. Please find our point-by-point responses below:

Editor: Thank you for submitting your manuscript to PLOS ONE. After careful consideration, we feel that it has merit but does not fully meet PLOS ONE’s publication criteria as it currently stands. Therefore, we invite you to submit a revised version of the manuscript that addresses all the points raised during the review process.

Response: Thank you for your time and consideration. The manuscript was revised to meet PLOS ONE’s publication criteria. The concerned limitation was discussed in detail in the Discussion section. The result interpretation, wording choices, and typos were revised throughout the manuscript.

Reviewer #1: The authors has addressed the concerns as best as they could without performing the requested additional testing results, which this reviewer still think is not satisfactory. Perhaps adding the rationale and explanation for having IgM and no IgG response in the tested case will provide additional scientific interest for the readers.

Response: Thank you for your suggestion. The rationale and explanation for having IgM and no IgG response in the tested case were added in the Discussion section.

Reviewer #3: 1) Although I agree with the flaws/limitations of the study as summarized by Reviewer 1, the study does add informative descriptive data for this region in Thailand despite the limitations.

Response: Thank you for your supportive comments. We believe that this study provides a useful piece of evidence for COVID-19 pandemic control from the region with a unique situation.

Reviewer #3: 2) The statement that females have higher IgM % than males in the Abstract and Results sections is not appropriate. Just provide the percentages and confidence intervals (CIs). As provided in the Results section, the CIs are nearly overlapping so it cannot be concluded that they are different. Please revise accordingly the statements below.

Abs: Females seemed to have higher IgM seropositive than male staff (1.0% vs. 0.5%).

Results: Female staff seemed to have higher rate of positive IgM (1.0%, 95% CI: 0.5–2.1%) than male (0.5%, 95% CI: 0.1–2.6%).

Response: Both statements were revised accordingly by providing only the percentages and confidence intervals (CIs) of each gender.

Reviewer #3: 3) In the conclusion: Consider revising FROM:

“Antibody testing should be promoted for mass screening, particularly in asymptomatic healthcare workers.”

TO:

“Accurate antibody testing is a valuable screening tool, particularly in asymptomatic healthcare workers.”

Response: The statement in the conclusion was revised accordingly.

Reviewer #3: 4) Typo:

Page 7: Only 1% of participants had previous negative PCR results while the rest never get tested (Table 1). … should be ‘got’ tested. Please read the paper again carefully and correct any grammatical errors.

Response: The typo on Page 7 was corrected. The manuscript was perused by the authors for any typo or grammatical error which was corrected accordingly.

We hope that our responses are satisfactory. Should there be anything that might improve our work, please kindly inform us. Thank you very much for your kind consideration.

Best Regards,

Assoc. Prof. Dr. Krit Pongpirul, MD, MPH, PhD.

On behalf of the authors

---

## [Decision Letter · Decision Letter 3]

5 Mar 2021

Seroprevalence of Hospital Staff in a Province with Zero COVID-19 Cases

PONE-D-20-25321R3

Dear Dr. Krit Pongpirul,

We’re pleased to inform you that your manuscript has been judged scientifically suitable for publication and will be formally accepted for publication once it meets all outstanding technical requirements.

Kind regards,

Daniela Flavia Hozbor

Academic Editor

PLOS ONE

Additional Editor Comments (optional):

Reviewers' comments:

Reviewer's Responses to Questions

**Comments to the Author**

1. If the authors have adequately addressed your comments raised in a previous round of review and you feel that this manuscript is now acceptable for publication, you may indicate that here to bypass the “Comments to the Author” section, enter your conflict of interest statement in the “Confidential to Editor” section, and submit your "Accept" recommendation.

Reviewer #3: All comments have been addressed

2. Is the manuscript technically sound, and do the data support the conclusions?

Reviewer #3: Yes

3. Has the statistical analysis been performed appropriately and rigorously? 

Reviewer #3: Yes

4. Have the authors made all data underlying the findings in their manuscript fully available?

Reviewer #3: Yes

5. Is the manuscript presented in an intelligible fashion and written in standard English?

Reviewer #3: Yes

6. Review Comments to the Author

Reviewer #3: The authors have responded adequately to all the points raised in my previous review. Thank you for responding.

7. PLOS authors have the option to publish the peer review history of their article (what does this mean?). If published, this will include your full peer review and any attached files.

Reviewer #3: No

---

## [Editor Report · Acceptance letter]

19 Mar 2021

PONE-D-20-25321R3 

Seroprevalence of hospital staff in a province with zero COVID-19 cases

Dear Dr. Pongpirul:

I'm pleased to inform you that your manuscript has been deemed suitable for publication in PLOS ONE. Congratulations! Your manuscript is now with our production department. 

Kind regards, 

on behalf of

Dr. Daniela Flavia Hozbor 

Academic Editor

PLOS ONE